# Tighten after Relax: Minimax-Optimal Sparse PCA in Polynomial Time

**Zhaoran Wang**    **Huanran Lu**    **Han Liu**
Department of Operations Research and Financial Engineering
Princeton University
Princeton, NJ 08540
{zhaoran,huanranl,hanliu}@princeton.edu

## Abstract

We provide statistical and computational analysis of sparse Principal Component Analysis (PCA) in high dimensions. The sparse PCA problem is highly nonconvex in nature. Consequently, though its global solution attains the optimal statistical rate of convergence, such solution is computationally intractable to obtain. Meanwhile, although its convex relaxations are tractable to compute, they yield estimators with suboptimal statistical rates of convergence. On the other hand, existing nonconvex optimization procedures, such as greedy methods, lack statistical guarantees.

In this paper, we propose a two-stage sparse PCA procedure that attains the optimal principal subspace estimator in polynomial time. The main stage employs a novel algorithm named sparse orthogonal iteration pursuit, which iteratively solves the underlying nonconvex problem. However, our analysis shows that this algorithm only has desired computational and statistical guarantees within a restricted region, namely the basin of attraction. To obtain the desired initial estimator that falls into this region, we solve a convex formulation of sparse PCA with early stopping.

Under an integrated analytic framework, we simultaneously characterize the computational and statistical performance of this two-stage procedure. Computationally, our procedure converges at the rate of $1/\sqrt{t}$ within the initialization stage, and at a geometric rate within the main stage. Statistically, the final principal subspace estimator achieves the minimax-optimal statistical rate of convergence with respect to the sparsity level $s^*$, dimension $d$ and sample size $n$. Our procedure motivates a general paradigm of tackling nonconvex statistical learning problems with provable statistical guarantees.

## 1 Introduction

We denote by $\boldsymbol{x}_1, \ldots, \boldsymbol{x}_n$ the $n$ realizations of a random vector $\boldsymbol{X} \in \mathbb{R}^d$ with population covariance matrix $\boldsymbol{\Sigma} \in \mathbb{R}^{d \times d}$. The goal of Principal Component Analysis (PCA) is to recover the top $k$ leading eigenvectors $\boldsymbol{u}_1^*, \ldots, \boldsymbol{u}_k^*$ of $\boldsymbol{\Sigma}$. In high dimensional settings with $d \gg n$, [1–3] showed that classical PCA can be inconsistent. Additional assumptions are needed to avoid such a curse of dimensionality. For example, when the first leading eigenvector is of primary interest, one common assumption is that $\boldsymbol{u}_1^*$ is sparse — the number of nonzero entries of $\boldsymbol{u}_1^*$, denoted by $s^*$, is smaller than $n$. Under such an assumption of sparsity, significant progress has been made on the methodological development [4–13] as well as theoretical understanding [1, 3, 14–21] of sparse PCA.

However, there remains a significant gap between the computational and statistical aspects of sparse PCA: No tractable algorithm is known to attain the statistical optimal sparse PCA estimator provably without relying on the spiked covariance assumption. This gap arises from the nonconvexity of sparse

PCA. In detail, the sparse PCA estimator for the first leading eigenvector $\boldsymbol{u}_1^*$ is

$$\widehat{\boldsymbol{u}}_1 = \underset{\|\boldsymbol{v}\|_2=1}{\operatorname{argmin}} -\boldsymbol{v}^T \widehat{\boldsymbol{\Sigma}} \boldsymbol{v}, \quad \text{subject to } \|\boldsymbol{v}\|_0 = s^*, \tag{1}$$

where $\widehat{\boldsymbol{\Sigma}}$ is the sample covariance estimator, $\|\cdot\|_2$ is the Euclidean norm, $\|\cdot\|_0$ gives the number of nonzero coordinates, and $s^*$ is the sparsity level of $\boldsymbol{u}_1^*$. Although this estimator has been proven to attain the optimal statistical rate of convergence [15, 17], its computation is intractable because it requires minimizing a concave function over cardinality constraints [22]. Estimating the top $k$ leading eigenvectors is even more challenging because of the extra orthogonality constraint on $\widehat{\boldsymbol{u}}_1, \ldots, \widehat{\boldsymbol{u}}_2$.

To address this computational issue, [5] proposed a convex relaxation approach, named DSPCA, for estimating the first leading eigenvector. [13] generalized DSPCA to estimate the principal subspace spanned by the top $k$ leading eigenvectors. Nevertheless, [13] proved the obtained estimator only attains the suboptimal $s^* \sqrt{\log d/n}$ statistical rate. Meanwhile, several methods have been proposed to directly address the underlying nonconvex problem (1), e.g., variants of power methods or iterative thresholding methods [10–12], greedy method [8], as well as regression-type methods [4, 6, 7, 18]. However, most of these methods lack statistical guarantees. There are several exceptions: (1) [11] proposed the truncated power method, which attains the optimal $\sqrt{s^* \log d/n}$ rate for estimating $\boldsymbol{u}_1^*$. However, it hinges on the assumption that the initial estimator $\boldsymbol{u}^{(0)}$ satisfies $\left|\sin \angle(\boldsymbol{u}^{(0)}, \boldsymbol{u}^*)\right| \le 1 - C$, where $C \in (0, 1)$ is a constant. Suppose $\boldsymbol{u}^{(0)}$ is chosen uniformly at random on the $\ell_2$ sphere, this assumption holds with probability decreasing to zero when $d \to \infty$ [23]. (2) [12] proposed an iterative thresholding method, which attains a near optimal statistical rate when estimating several individual leading eigenvectors. [18] proposed a regression-type method, which attains the optimal principal subspace estimator. However, these two methods hinge on the spiked covariance assumption, and require the data to be exactly Gaussian (sub-Gaussian not included). For them, the spiked covariance assumption is crucial, because they use diagonal thresholding method [1] to obtain the initialization, which would fail when the assumption of spiked covariance doesn't hold, or each coordinate of $\boldsymbol{X}$ has the same variance. Besides, except [12] and [18], all the computational procedures only recover the first leading eigenvector, and leverage the deflation method [24] to recover the rest, which leads to identifiability and orthogonality issues when the top $k$ eigenvalues of $\boldsymbol{\Sigma}$ are not distinct.

To close the gap between computational and statistical aspects of sparse PCA, we propose a two-stage procedure for estimating the $k$-dimensional principal subspace $\mathcal{U}^*$ spanned by the top $k$ leading eigenvectors $\boldsymbol{u}_1^*, \ldots, \boldsymbol{u}_k^*$. The details of the two stages are as follows: (1) For the main stage, we propose a novel algorithm, named sparse orthogonal iteration pursuit, to directly estimate the principal subspace of $\boldsymbol{\Sigma}$. Our analysis shows, when its initialization falls into a restricted region, namely the basin of attraction, this algorithm enjoys fast optimization rate of convergence, and attains the optimal principal subspace estimator. (2) To obtain the desired initialization, we compute a convex relaxation of sparse PCA. Unlike [5, 13], which calculate the exact minimizers, we early stop the corresponding optimization algorithm as soon as the iterative sequence enters the basin of attraction for the main stage. The rationale is, this convex optimization algorithm converges at a slow sublinear rate towards a suboptimal estimator, and incurs relatively high computational overhead within each iteration.

Under a unified analytic framework, we provide simultaneous statistical and computational guarantees for this two-stage procedure. Given the sample size $n$ is sufficiently large, and the eigengap between the $k$-th and $(k + 1)$-th eigenvalues of the population covariance matrix $\boldsymbol{\Sigma}$ is nonzero, we prove: (1) The final subspace estimator $\widehat{\mathcal{U}}$ attained by our two-stage procedure achieves the minimax-optimal $\sqrt{s^* \log d/n}$ statistical rate of convergence. (2) Within the initialization stage, the iterative sequence of subspace estimators $\{\mathcal{U}^{(t)}\}_{t=0}^{T}$ (at the $T$-th iteration we early stop the initialization stage) satisfies

$$D(\mathcal{U}^*, \mathcal{U}^{(t)}) \le \underbrace{\delta_1(\boldsymbol{\Sigma}) \cdot s^* \sqrt{\log d/n}}_{\text{Statistical Error}} + \underbrace{\delta_2(k, s^*, d, n) \cdot 1/\sqrt{t}}_{\text{Optimization Error}} \tag{2}$$

with high probability. Here $D(\cdot, \cdot)$ is the subspace distance, while $s^*$ is the sparsity level of $\mathcal{U}^*$, both of which will be defined in §2. Here $\delta_1(\boldsymbol{\Sigma})$ is a quantity which depends on the population covariance matrix $\boldsymbol{\Sigma}$, while $\delta_2(k, s^*, d, n)$ depends on $k$, $s^*$, $d$ and $n$ (see §4 for details). (3) Within the main stage, the iterative sequence $\{\mathcal{U}^{(t)}\}_{t=T+1}^{T+\widetilde{T}}$ (where $\widetilde{T}$ denotes the total number of iterations of sparse

orthogonal iteration pursuit) satisfies

$$D\big(\mathcal{U}^*,\mathcal{U}^{(t)}\big) \le \underbrace{\delta_3(\boldsymbol{\Sigma},k)\cdot \overbrace{\sqrt{s^*\log d/n}}^{\text{Optimal Rate}}}_{\text{Statistical Error}} + \underbrace{\gamma(\boldsymbol{\Sigma})^{(t-T-1)/4}\cdot D\big(\mathcal{U}^*,\mathcal{U}^{(T+1)}\big)}_{\text{Optimization Error}} \qquad (3)$$

with high probability, where $\delta_3(\boldsymbol{\Sigma},k)$ is a quantity that only depends on $\boldsymbol{\Sigma}$ and $k$, and

$$\gamma(\boldsymbol{\Sigma}) = [3\lambda_{k+1}(\boldsymbol{\Sigma}) + \lambda_k(\boldsymbol{\Sigma})]/[\lambda_{k+1}(\boldsymbol{\Sigma}) + 3\lambda_k(\boldsymbol{\Sigma})] < 1. \qquad (4)$$

Here $\lambda_k(\boldsymbol{\Sigma})$ and $\lambda_{k+1}(\boldsymbol{\Sigma})$ are the $k$-th and $(k+1)$-th eigenvalues of $\boldsymbol{\Sigma}$. See §4 for more details.

Unlike previous works, our theory and method don't depend on the spiked covariance assumption, or require the data distribution to be Gaussian.

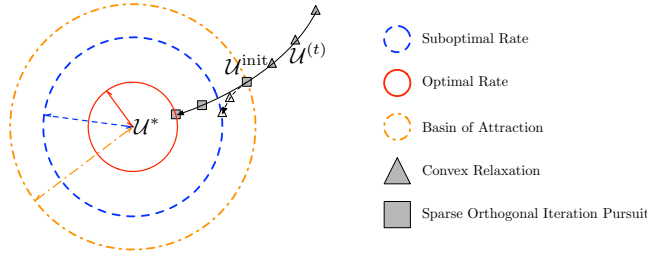

Figure 1: An illustration of our two-stage procedure.

Our analysis shows, at the initialization stage, the optimization error decays to zero at the rate of $1/\sqrt{t}$. However, the upper bound of $D\big(\mathcal{U}^*,\mathcal{U}^{(t)}\big)$ in (2) can't be smaller than the suboptimal $s^*\sqrt{\log d/n}$ rate of convergence, even with infinite number of iterations. This phenomenon, which is illustrated in Figure 1, reveals the limit of the convex relaxation approaches for sparse PCA. Within the main stage, as the optimization error term in (3) decreases to zero geometrically, the upper bound of $D\big(\mathcal{U}^*,\mathcal{U}^{(t)}\big)$ decreases towards the $\sqrt{s^*\log d/n}$ statistical rate of convergence, which is minimax-optimal with respect to the sparsity level $s^*$, dimension $d$ and sample size $n$ [17]. Moreover, in Theorem 2 we will show that, the basin of attraction for the proposed sparse orthogonal iteration pursuit algorithm can be characterized as

$$\mathcal{U}: D\big(\mathcal{U}^*,\mathcal{U}\big) \le R = \min\left\{\sqrt{k\gamma(\boldsymbol{\Sigma})\left[1 - \gamma(\boldsymbol{\Sigma})^{1/2}\right]/2}, \sqrt{2\gamma(\boldsymbol{\Sigma})}/4\right\}. \qquad (5)$$

Here $\gamma(\boldsymbol{\Sigma})$ is defined in (4) and $R$ denotes its radius.

The contribution of this paper is three-fold: (1) We propose the first tractable procedure that provably attains the subspace estimator with minimax-optimal statistical rate of convergence with respect to the sparsity level $s^*$, dimension $d$ and sample size $n$, without relying on the restrictive spiked covariance assumption or the Gaussian assumption. (2) We propose a novel algorithm named sparse orthogonal iteration pursuit, which converges to the optimal estimator at a fast geometric rate. The computation within each iteration is highly efficient compared with convex relaxation approaches. (3) We build a joint analytic framework that simultaneously captures the computational and statistical properties of sparse PCA. Under this framework, we characterize the phenomenon of basin of attraction for the proposed sparse orthogonal iteration pursuit algorithm. In comparison with our previous work on nonconvex $M$-estimators [25], our analysis provides a more general paradigm of solving nonconvex learning problems with provable guarantees. One byproduct of our analysis is novel techniques for analyzing the statistical properties of the intermediate solutions of the Alternating Direction Method of Multipliers [26].

**Notation:** Let $\mathbf{A} = [\mathbf{A}_{i,j}] \in \mathbb{R}^{d\times d}$ and $\boldsymbol{v} = (v_1,\ldots,v_d)^T \in \mathbb{R}^d$. The $\ell_q$ norm $(q \ge 1)$ of $\boldsymbol{v}$ is $\|\boldsymbol{v}\|_q$. Specifically, $\|\boldsymbol{v}\|_0$ gives the number of nonzero entries of $\boldsymbol{v}$. For matrix $\mathbf{A}$, the $i$-th largest eigenvalue and singular value are $\lambda_i(\mathbf{A})$ and $\sigma_i(\mathbf{A})$. For $q \ge 1$, $\|\mathbf{A}\|_q$ is the matrix operator $q$-norm, e.g., we have $\|\mathbf{A}\|_2 = \sigma_1(\mathbf{A})$. The Frobenius norm is denoted as $\|\mathbf{A}\|_F$. For $\mathbf{A}_1$ and $\mathbf{A}_2$, their inner product is $\langle \mathbf{A}_1, \mathbf{A}_2 \rangle = \operatorname{tr}(\mathbf{A}_1^T \mathbf{A}_2)$. For a set $\mathcal{S}$, $|\mathcal{S}|$ denotes its cardinality. The $d \times d$ identity matrix is $\mathbf{I}_d$.

For index sets $\mathcal{I}, \mathcal{J} \subseteq \{1, \ldots, d\}$, we define $\mathbf{A}_{\mathcal{I},\mathcal{J}} \in \mathbb{R}^{d \times d}$ to be the matrix whose $(i,j)$-th entry is $\mathbf{A}_{i,j}$ if $i \in \mathcal{I}$ and $j \in \mathcal{J}$, and zero otherwise. When $\mathcal{I} = \mathcal{J}$, we abbreviate it as $\mathbf{A}_{\mathcal{I}}$. If $\mathcal{I}$ or $\mathcal{J}$ is $\{1, \ldots, d\}$, we replace it with a dot, e.g., $\mathbf{A}_{\mathcal{I},\cdot}$. We denote by $\mathbf{A}_{i,*} \in \mathbb{R}^d$ the $i$-th row vector of $\mathbf{A}$. A matrix is orthonormal if its columns are unit length orthogonal vectors. The $(p,q)$-norm of a matrix, denoted as $\|\mathbf{A}\|_{p,q}$, is obtained by first taking the $\ell_p$ norm of each row, and then taking $\ell_q$ norm of these row norms. We denote $\text{diag}(\mathbf{A})$ to be the vector consisting of the diagonal entries of $\mathbf{A}$. With a little abuse of notation, we denote by $\text{diag}(\boldsymbol{v})$ the the diagonal matrix with $v_1, \ldots, v_d$ on its diagonal. Hereafter, we use generic numerical constants $C, C', C'', \ldots$, whose values change from line to line.

## 2 Background

In the following, we introduce the distance between subspaces and the notion of sparsity for subspace.

**Subspace Distance:** Let $\mathcal{U}$ and $\mathcal{U}'$ be two $k$-dimensional subspaces of $\mathbb{R}^d$. We denote the projection matrix onto them by $\boldsymbol{\Pi}$ and $\boldsymbol{\Pi}'$ respectively. One definition of the distance between $\mathcal{U}$ and $\mathcal{U}'$ is

$$D(\mathcal{U}, \mathcal{U}') = \|\boldsymbol{\Pi} - \boldsymbol{\Pi}'\|_F. \tag{6}$$

This definition is invariant to the rotations of the orthonormal basis.

**Subspace Sparsity:** For the $k$-dimensional principal subspace $\mathcal{U}^*$ of $\boldsymbol{\Sigma}$, the definition of its sparsity should be invariant to the choice of basis, because $\boldsymbol{\Sigma}$'s top $k$ eigenvalues might be not distinct. Here we define the sparsity level $s^*$ of $\mathcal{U}^*$ to be the number of nonzero coefficients on the diagonal of its projection matrix $\boldsymbol{\Pi}^*$. One can verify that (see [17] for details)

$$s^* = \big|\text{supp}[\text{diag}(\boldsymbol{\Pi}^*)]\big| = \|\mathbf{U}^*\|_{2,0}, \tag{7}$$

where $\|\cdot\|_{2,0}$ gives the row-sparsity level, i.e., the number of nonzero rows. Here the columns of $\mathbf{U}^*$ can be any orthonormal basis of $\mathcal{U}^*$. This definition reduces to the sparsity of $\boldsymbol{u}_1^*$ when $k = 1$.

**Subspace Estimation:** For the $k$-dimensional $s^*$-sparse principal subspace $\mathcal{U}^*$ of $\boldsymbol{\Sigma}$, [17] considered the following estimator for the orthonormal matrix $\mathbf{U}^*$ consisting of the basis of $\mathcal{U}^*$,

$$\widehat{\mathbf{U}} = \underset{\mathbf{U} \in \mathbb{R}^{d \times k}}{\text{argmin}} -\big\langle \widehat{\boldsymbol{\Sigma}}, \mathbf{U}\mathbf{U}^T \big\rangle, \quad \text{subject to } \mathbf{U} \text{ orthonormal, and } \|\mathbf{U}\|_{2,0} \le s^*, \tag{8}$$

where $\widehat{\boldsymbol{\Sigma}}$ is an estimator of $\boldsymbol{\Sigma}$. Let $\widehat{\mathcal{U}}$ be the column space of $\widehat{\mathbf{U}}$. [17] proved that, assuming $\widehat{\boldsymbol{\Sigma}}$ is the sample covariance estimator, and the data are independent sub-Gaussian, $\widehat{\mathcal{U}}$ attains the optimal statistical rate. However, direct computation of this estimator is NP-hard even for $k = 1$ [22].

## 3 A Two-stage Procedure for Sparse PCA

In this following, we present the two-stage procedure for sparse PCA. We will first introduce sparse orthogonal iteration pursuit for the main stage and then present the convex relaxation for initialization.

---

**Algorithm 1** Main stage: Sparse orthogonal iteration pursuit. Here $T$ denotes the total number of iterations of the initialization stage. To unify the later analysis, let $t$ start from $T + 1$.

---

1: **Function**: $\widehat{\mathbf{U}} \leftarrow \text{Sparse\_Orthogonal\_Iteration\_Pursuit}\big(\widehat{\boldsymbol{\Sigma}}, \mathbf{U}^{\text{init}}\big)$
2: **Input:** Covariance Matrix Estimator $\widehat{\boldsymbol{\Sigma}}$, Initialization $\mathbf{U}^{\text{init}}$
3: **Parameter:** Sparsity Parameter $\widehat{s}$, Maximum Number of Iterations $\widetilde{T}$
4: **Initialization:** $\widetilde{\mathbf{U}}^{(T+1)} \leftarrow \text{Truncate}\big(\mathbf{U}^{\text{init}}, \widehat{s}\big)$, $\quad \mathbf{U}^{(T+1)}, \mathbf{R}_2^{(T+1)} \leftarrow \text{Thin\_QR}\big(\widetilde{\mathbf{U}}^{(T+1)}\big)$
5: **For** $t = T + 1, \ldots, T + \widetilde{T} - 1$
6: $\quad \widetilde{\mathbf{V}}^{(t+1)} \leftarrow \widehat{\boldsymbol{\Sigma}} \cdot \mathbf{U}^{(t)}, \qquad\qquad \mathbf{V}^{(t+1)}, \mathbf{R}_1^{(t+1)} \leftarrow \text{Thin\_QR}\big(\widetilde{\mathbf{V}}^{(t+1)}\big)$
7: $\quad \widetilde{\mathbf{U}}^{(t+1)} \leftarrow \text{Truncate}\big(\mathbf{V}^{(t+1)}, \widehat{s}\big), \quad \mathbf{U}^{(t+1)}, \mathbf{R}_2^{(t+1)} \leftarrow \text{Thin\_QR}\big(\widetilde{\mathbf{U}}^{(t+1)}\big)$
8: **End For**
9: **Output:** $\widehat{\mathbf{U}} \leftarrow \mathbf{U}^{(T+\widetilde{T})}$

---

**Sparse Orthogonal Iteration Pursuit:** For the main stage, we propose sparse orthogonal iteration pursuit (Algorithm 1) to solve (8). In Algorithm 1, $\mathsf{Truncate}(\cdot,\cdot)$ (Line 7) is defined in Algorithm 2. In Lines 6 and 7, $\mathsf{Thin\_QR}(\cdot)$ denotes the thin QR decomposition (see [27] for details). In detail, $\mathbf{V}^{(t+1)} \in \mathbb{R}^{d\times k}$ and $\mathbf{U}^{(t+1)} \in \mathbb{R}^{d\times k}$ are orthonormal matrices, and they satisfy $\mathbf{V}^{(t+1)} \cdot \mathbf{R}_1^{(t+1)} = \widetilde{\mathbf{V}}^{(t+1)}$, and $\mathbf{U}^{(t+1)} \cdot \mathbf{R}_2^{(t+1)} = \widetilde{\mathbf{U}}^{(t+1)}$, where $\mathbf{R}_1^{(t+1)}, \mathbf{R}_2^{(t+1)} \in \mathbb{R}^{k\times k}$. This decomposition can be accomplished with $\mathcal{O}(k^2 d)$ operations using Householder algorithm [27]. Here remind that $k$ is the rank of the principal subspace of interest, which is much smaller than the dimension $d$.

Algorithm 1 consists of two steps: (1) Line 6 performs a matrix multiplication and a renormalization using QR decomposition. This step is named orthogonal iteration in numerical analysis [27]. When the first leading eigenvector ($k = 1$) is of interest, it reduces to the well-known power iteration. The intuition behind this step can be understood as follows. We consider the minimization problem in (8) without the row-sparsity constraint. Note that the gradient of the objective function is $-2\widehat{\boldsymbol{\Sigma}} \cdot \mathbf{U}^{(t)}$. Hence, the gradient descent update scheme for this problem is

$$\widetilde{\mathbf{V}}^{(t+1)} \leftarrow \mathcal{P}_{\mathrm{orth}}\big(\mathbf{U}^{(t)} + \eta \cdot 2\widehat{\boldsymbol{\Sigma}} \cdot \mathbf{U}^{(t)}\big), \tag{9}$$

where $\eta$ is the step size, and $\mathcal{P}_{\mathrm{orth}}(\cdot)$ denotes the renormalization step. [28] showed that the optimal step size $\eta$ is infinity. Thus we have $\mathcal{P}_{\mathrm{orth}}\big(\mathbf{U}^{(t)} + \eta \cdot 2\widehat{\boldsymbol{\Sigma}} \cdot \mathbf{U}^{(t)}\big) = \mathcal{P}_{\mathrm{orth}}\big(\eta \cdot 2\widehat{\boldsymbol{\Sigma}} \cdot \mathbf{U}^{(t)}\big) = \mathcal{P}_{\mathrm{orth}}\big(\widehat{\boldsymbol{\Sigma}} \cdot \mathbf{U}^{(t)}\big)$, which implies that (9) is equivalent to Line 6. (2) In Line 7, we take a truncation step to enforce the row-sparsity constraint in (8). In detail, we greedily select the $\widehat{s}$ most important rows. To enforce the orthonormality constraint in (8), we perform another renormalization step after the truncation. Note that the QR decomposition in Line 7 gives a both orthonormal and row-sparse $\mathbf{U}^{(t+1)}$, because $\widetilde{\mathbf{U}}^{(t+1)}$ is row-sparse by truncation, and QR decomposition preserves its row-sparsity. By iteratively performing these two steps, we are approximately solving the nonconvex problem in (8). Although it is not clear whether this procedure achieves the global minimum of (8), we will prove that, the obtained estimator enjoys the same optimal statistical rate of convergence as the global minimum.

---

**Algorithm 2** Main stage: The $\mathsf{Truncate}(\cdot,\cdot)$ function used in Line 7 of Algorithm 1.

1: **Function**: $\widetilde{\mathbf{U}}^{(t+1)} \leftarrow \mathsf{Truncate}\big(\mathbf{V}^{(t+1)}, \widehat{s}\big)$
2: **Row Sorting**: $\mathcal{I}_{\widehat{s}} \leftarrow$ The set of row index $i's$ with the top $\widehat{s}$ largest $\big\|\mathbf{V}_{i,*}^{(t+1)}\big\|_2$'s
3: **Truncation**: $\widetilde{\mathbf{U}}_{i,*}^{(t+1)} \leftarrow \mathbb{1}\big(i \in \mathcal{I}_{\widehat{s}}\big) \cdot \mathbf{V}_{i,*}^{(t+1)}, \quad$ for all $i \in \{1,\ldots,d\}$
4: **Output**: $\widetilde{\mathbf{U}}^{(t+1)}$

---

**Algorithm 3** Initialization stage: Solving convex relaxation (10) using ADMM. In Lines 6 and 7, we need to solve two subproblems. The first one is equivalent to projecting $\boldsymbol{\Phi}^{(t)} - \boldsymbol{\Theta}^{(t)} + \widehat{\boldsymbol{\Sigma}}/\rho$ to $\mathcal{A}$. This projection can be computed using Algorithm 4 in [29]. The second can be solved by entry-wise soft-thresholding shown in Algorithm 5 in [29]. We defer these two algorithms and their derivations to the extended version [29] of this paper.

1: **Function**: $\mathbf{U}^{\mathrm{init}} \leftarrow \mathsf{ADMM}\big(\widehat{\boldsymbol{\Sigma}}\big)$
2: **Input**: Covariance Matrix Estimator $\widehat{\boldsymbol{\Sigma}}$
3: **Parameter**: Regularization Parameter $\rho > 0$ in (10), Penalty Parameter $\beta > 0$ of the Augmented Lagrangian, Maximum Number of Iterations $T$
4: $\boldsymbol{\Pi}^{(0)} \leftarrow \mathbf{0}, \boldsymbol{\Phi}^{(0)} \leftarrow \mathbf{0}, \boldsymbol{\Theta}^{(0)} \leftarrow \mathbf{0}$
5: **For** $t = 0,\ldots,T-1$
6: $\quad \boldsymbol{\Pi}^{(t+1)} \leftarrow \mathrm{argmin}\big\{L\big(\boldsymbol{\Pi}, \boldsymbol{\Phi}^{(t)}, \boldsymbol{\Theta}^{(t)}\big) + \beta/2 \cdot \big\|\boldsymbol{\Pi} - \boldsymbol{\Phi}^{(t)}\big\|_F^2 \mid \boldsymbol{\Pi} \in \mathcal{A}\big\}$
7: $\quad \boldsymbol{\Phi}^{(t+1)} \leftarrow \mathrm{argmin}\big\{L\big(\boldsymbol{\Pi}^{(t+1)}, \boldsymbol{\Phi}, \boldsymbol{\Theta}^{(t)}\big) + \beta/2 \cdot \big\|\boldsymbol{\Pi}^{(t+1)} - \boldsymbol{\Phi}\big\|_F^2 \mid \boldsymbol{\Phi} \in \mathcal{B}\big\}$
8: $\quad \boldsymbol{\Theta}^{(t+1)} \leftarrow \boldsymbol{\Theta}^{(t)} - \beta\big(\boldsymbol{\Pi}^{(t+1)} - \boldsymbol{\Phi}^{(t+1)}\big)$
9: **End For**
10: $\overline{\boldsymbol{\Pi}}^{(T)} \leftarrow 1/T \cdot \sum_{t=0}^{T} \boldsymbol{\Pi}^{(t)}$, let the columns of $\mathbf{U}^{\mathrm{init}}$ be the top $k$ leading eigenvectors of $\overline{\boldsymbol{\Pi}}^{(T)}$
11: **Output**: $\mathbf{U}^{\mathrm{init}} \in \mathbb{R}^{d\times k}$

---

**Convex Relaxation for Initialization:** To obtain a good initialization for sparse orthogonal iteration pursuit, we consider the following convex minimization problem proposed by [5, 13]

$$\text{minimize} \left\{ -\langle \widehat{\boldsymbol{\Sigma}}, \boldsymbol{\Pi} \rangle + \rho \|\boldsymbol{\Pi}\|_{1,1} \mid \text{tr}(\boldsymbol{\Pi}) = k, \ \mathbf{0} \preceq \boldsymbol{\Pi} \preceq \mathbf{I}_d \right\}, \tag{10}$$

which relaxes the combinatorial optimization problem in (8). The intuition behind this relaxation can be understood as follows: (1) $\boldsymbol{\Pi}$ is a reparametrization for $\mathbf{U}\mathbf{U}^T$ in (8), which is a projection matrix with $k$ nonzero eigenvalues of 1. In (10), this constraint is relaxed to $\text{tr}(\boldsymbol{\Pi}) = k$ and $\mathbf{0} \preceq \boldsymbol{\Pi} \preceq \mathbf{I}_d$, which indicates that the eigenvalues of $\boldsymbol{\Pi}$ should be in $[0, 1]$ while the sum of them is $k$. (2) For the row-sparsity constraint in (8), [13] proved that $\|\boldsymbol{\Pi}^*\|_{0,0} \leq |\text{supp}[\text{diag}(\boldsymbol{\Pi}^*)]|^2 = \|\mathbf{U}^*\|_{2,0}^2 = (s^*)^2$. Correspondingly, the row-sparsity constraint in (8) translates to $\|\boldsymbol{\Pi}\|_{0,0} \leq (s^*)^2$, which is relaxed to the regularization term $\|\boldsymbol{\Pi}\|_{1,1}$ in (10). For notational simplicity, we define

$$\mathcal{A} = \left\{ \boldsymbol{\Pi} \colon \boldsymbol{\Pi} \in \mathbb{R}^{d \times d}, \ \text{tr}(\boldsymbol{\Pi}) = k, \ \mathbf{0} \preceq \boldsymbol{\Pi} \preceq \mathbf{I}_d \right\}. \tag{11}$$

Note (10) has both nonsmooth regularization term and nontrivial constraint $\mathcal{A}$. We use the Alternating Direction Method of Multipliers (ADMM, Algorithm 3). It considers the equivalent form of (10)

$$\text{minimize} \left\{ -\langle \widehat{\boldsymbol{\Sigma}}, \boldsymbol{\Pi} \rangle + \rho \|\boldsymbol{\Phi}\|_{1,1} \mid \boldsymbol{\Pi} = \boldsymbol{\Phi}, \ \boldsymbol{\Pi} \in \mathcal{A}, \ \boldsymbol{\Phi} \in \mathcal{B} \right\}, \quad \text{where } \mathcal{B} = \mathbb{R}^{d \times d}, \tag{12}$$

and iteratively minimizes the augmented Lagrangian $L(\boldsymbol{\Pi}, \boldsymbol{\Phi}, \boldsymbol{\Theta}) + \beta/2 \cdot \|\boldsymbol{\Pi} - \boldsymbol{\Phi}\|_F^2$, where

$$L(\boldsymbol{\Pi}, \boldsymbol{\Phi}, \boldsymbol{\Theta}) = -\langle \widehat{\boldsymbol{\Sigma}}, \boldsymbol{\Pi} \rangle + \rho \|\boldsymbol{\Phi}\|_{1,1} - \langle \boldsymbol{\Theta}, \boldsymbol{\Pi} - \boldsymbol{\Phi} \rangle, \quad \boldsymbol{\Pi} \in \mathcal{A}, \ \boldsymbol{\Phi} \in \mathcal{B}, \ \boldsymbol{\Theta} \in \mathbb{R}^{d \times d} \tag{13}$$

is the Lagrangian corresponding to (12), $\boldsymbol{\Theta} \in \mathbb{R}^{d \times d}$ is the Lagrange multiplier associated with the equality constraint $\boldsymbol{\Pi} = \boldsymbol{\Phi}$, and $\beta > 0$ is a penalty parameter that enforces such an equality constraint. Note that other variants of ADMM, e.g., Peaceman-Rachford Splitting Method [30] is also applicable, which would yield similar theoretical guarantees along with improved practical performance.

## 4 Theoretical Results

To describe our results, we define the model class $\mathcal{M}_d(\boldsymbol{\Sigma}, k, s^*)$ as follows,

$$\mathcal{M}_d(\boldsymbol{\Sigma}, k, s^*) \colon \begin{cases} \boldsymbol{X} = \boldsymbol{\Sigma}^{1/2} \boldsymbol{Z}, \text{ where } \boldsymbol{Z} \in \mathbb{R}^d \text{ is sub-Gaussian with mean zero,} \\ \qquad \text{variance proxy less than 1, and covariance matrix } \mathbf{I}_d; \\ \text{The } k\text{-dimensional principal subspace } \mathcal{U}^* \text{ of } \boldsymbol{\Sigma} \text{ is } s^*\text{-sparse;} \ \lambda_k(\boldsymbol{\Sigma}) - \lambda_{k+1}(\boldsymbol{\Sigma}) > 0. \end{cases}$$

where $\boldsymbol{\Sigma}^{1/2}$ satisfies $\boldsymbol{\Sigma}^{1/2} \cdot \boldsymbol{\Sigma}^{1/2} = \boldsymbol{\Sigma}$. Here remind the sparsity of $\mathcal{U}^*$ is defined in (7) and $\lambda_j(\boldsymbol{\Sigma})$ is the $j$-th eigenvalue of $\boldsymbol{\Sigma}$. For notational simplicity, hereafter we abbreviate $\lambda_j(\boldsymbol{\Sigma})$ as $\lambda_j$. This model class doesn't restrict $\boldsymbol{\Sigma}$ to spiked covariance matrices, where the $(k+1), \ldots, d$-th eigenvalues of $\boldsymbol{\Sigma}$ can only be identical. Moreover, we don't require $\boldsymbol{X}$ to be exactly Gaussian, which is a crucial requirement in several previous works, e.g., [12, 18].

We first introduce some notation. Remind $D(\cdot, \cdot)$ is the subspace distance defined in (6). Note that $\gamma(\boldsymbol{\Sigma}) < 1$ is defined in (4) and will be abbreviated as $\gamma$ hereafter. We define

$$n_{\min} = C \cdot (s^*)^2 \log d \cdot \min \left\{ \sqrt{k \cdot \gamma(1 - \gamma^{1/2})/2}, \ \sqrt{2\gamma}/4 \right\}^2 \cdot (\lambda_k - \lambda_{k+1})^2 / \lambda_1^2, \tag{14}$$

which denotes the required sample complexity. We also define

$$\zeta_1 = [C\lambda_1/(\lambda_k - \lambda_{k+1})] \cdot s^* \sqrt{\log d/n}, \quad \zeta_2 = \left[ 4/\sqrt{\lambda_k - \lambda_{k+1}} \right] \cdot \left( k \cdot s^* \cdot d^2 \log d/n \right)^{1/4}, \tag{15}$$

which will be used in the analysis of the first stage, and

$$\xi_1 = C\sqrt{k} \cdot [\lambda_k/(\lambda_k - \lambda_{k+1})]^2 \cdot \left[ \sqrt{\lambda_1 \lambda_{k+1}}/(\lambda_k - \lambda_{k+1}) \right] \cdot \sqrt{s^* \cdot (k + \log d)/n}, \tag{16}$$

which will be used in the analysis of the main stage. Meanwhile, remind the radius of the basin of attraction for sparse orthogonal iteration pursuit is defined in (5). We define

$$T_{\min} = \left\lceil \zeta_2^2/(R - \zeta_1)^2 \right\rceil, \qquad \widetilde{T}_{\min} = 4 \left\lceil \log(R/\xi_1)/\log(1/\gamma) \right\rceil \tag{17}$$

as the required minimum numbers of iterations of the two stages respectively. The following results will be proved in the extended version [29] of this paper accordingly.

**Main Result:** Recall that $\mathcal{U}^{(t)}$ denotes the subspace spanned by the columns of $\mathbf{U}^{(t)}$ in Algorithm 1.

**Theorem 1.** Let $\boldsymbol{x}_1, \ldots, \boldsymbol{x}_n$ be independent realizations of $\boldsymbol{X} \in \mathcal{M}_d(\boldsymbol{\Sigma}, k, s^*)$ with $n \geq n_{\min}$, and $\widehat{\boldsymbol{\Sigma}}$ be the sample covariance matrix. Suppose the regularization parameter $\rho = C\lambda_1\sqrt{\log d/n}$ for a sufficiently large $C > 0$ in (10) and the penalty parameter $\beta$ of ADMM (Line 3 of Algorithm 3) is $\beta = d \cdot \rho / \sqrt{k}$. Also, suppose the sparsity parameter $\widehat{s}$ in Algorithm 1 (Line 3) is chosen such that $\widehat{s} = C \max \left\{ \lceil 4k/(\gamma^{-1/2} - 1)^2 \rceil, 1 \right\} \cdot s^*$, where $C \geq 1$ is an integer constant. After $T \geq T_{\min}$ iterations of Algorithm 3 and then $\widetilde{T} \geq \widetilde{T}_{\min}$ iterations of Algorithm 1, we obtain $\widehat{\mathcal{U}} = \mathcal{U}^{(T+\widetilde{T})}$ and

$$D\big(\mathcal{U}^*, \widehat{\mathcal{U}}\big) \leq C\xi_1 = C'\sqrt{k} \cdot [\lambda_k/(\lambda_k - \lambda_{k+1})]^2 \cdot \left[\sqrt{\lambda_1\lambda_{k+1}}/(\lambda_k - \lambda_{k+1})\right] \cdot \sqrt{s^* \cdot (k + \log d)/n}$$

with high probability. Here the equality follows from the definition of $\xi_1$ in (16).

**Minimax-Optimality:** To establish the optimality of Theorem 1, we consider a smaller model class $\widetilde{\mathcal{M}}_d(\boldsymbol{\Sigma}, k, s^*, \kappa)$, which is the same as $\mathcal{M}_d(\boldsymbol{\Sigma}, k, s^*)$ except the eigengap of $\boldsymbol{\Sigma}$ satisfies $\lambda_k - \lambda_{k+1} > \kappa\lambda_k$ for some constant $\kappa > 0$. This condition is mild compared to previous works, e.g., [12] assumes $\lambda_k - \lambda_{k+1} \geq \kappa\lambda_1$, which is more restrictive because $\lambda_1 \geq \lambda_k$. Within $\widetilde{\mathcal{M}}$, we assume that the rank $k$ of the principal subspace is fixed. This assumption is reasonable, e.g., in applications like population genetics [31], the rank $k$ of principal subspaces represents the number of population groups, which doesn't increase when the sparsity level $s^*$, dimension $d$ and sample size $n$ are growing.

Theorem 3.1 of [17] implies the following minimax lower bound

$$\inf_{\widetilde{\mathcal{U}}} \sup_{\boldsymbol{X} \in \widetilde{\mathcal{M}}_d(\boldsymbol{\Sigma}, k, s^*)} \mathbb{E}\, D\big(\widetilde{\mathcal{U}}, \mathcal{U}^*\big)^2 \geq C\lambda_1\lambda_{k+1}/(\lambda_k - \lambda_{k+1})^2 \cdot (s^* - k) \cdot \left\{k + \log[(d-k)/(s^*-k)]\right\}/n,$$

where $\widetilde{\mathcal{U}}$ denotes any principal subspace estimator. Suppose $s^*$ and $d$ are sufficiently large (to avoid trivial cases), the right-hand side is lower bounded by $C'\lambda_1\lambda_{k+1}/(\lambda_k - \lambda_{k+1})^2 \cdot s^* \cdot (k + 1/4 \cdot \log d)/n$. By Lemma 2.1 in [29], we have $D\big(\mathcal{U}^*, \widehat{\mathcal{U}}\big) \leq \sqrt{2k}$. For $n$, $d$ and $s^*$ sufficiently large, it is easy to derive the same upper bound in expectation from in Theorem 1. It attains the minimax lower bound above within $\widetilde{\mathcal{M}}_d(\boldsymbol{\Sigma}, k, s^*, \kappa)$, up to the $1/4$ constant in front of $\log d$ and a total constant of $k \cdot \kappa^{-4}$.

**Analysis of the Main Stage:** Remind that $\mathcal{U}^{(t)}$ is the subspace spanned by the columns of $\mathbf{U}^{(t)}$ in Algorithm 1, and the initialization is $\mathbf{U}^{\text{init}}$ while its column space is $\mathcal{U}^{\text{init}}$.

**Theorem 2.** Under the same condition as in Theorem 1, and provided that $D\big(\mathcal{U}^*, \mathcal{U}^{\text{init}}\big) \leq R$, the iterative sequence $\mathcal{U}^{(T+1)}, \mathcal{U}^{(T+2)}, \ldots, \mathcal{U}^{(t)}, \ldots$ satisfies

$$D\big(\mathcal{U}^*, \mathcal{U}^{(t)}\big) \leq \underbrace{C\xi_1}_{\text{Statistical Error}} + \underbrace{\gamma^{(t-T-1)/4} \cdot \gamma^{-1/2}R}_{\text{Optimization Error}} \qquad (18)$$

with high probability, where $\xi_1$ is defined in (16), $R$ is defined in (5), and $\gamma$ is defined in (4).

Theorem 2 shows that, as long as $\mathcal{U}^{\text{init}}$ falls into its basin of attraction, sparse orthogonal iteration pursuit converges at a geometric rate of convergence in optimization error since $\gamma < 1$. According to the definition of $\gamma$ in (4), when $\lambda_k$ is close to $\lambda_{k+1}$, $\gamma$ is close to 1, then the optimization error term decays at a slower rate. Here the optimization error doesn't increase with dimension $d$, which makes this algorithm suitable to solve ultra high dimensional problems. In (18), when $t$ is sufficiently large such that $\gamma^{(t-T-1)/4} \cdot \gamma^{-1/2}R \leq \xi_1$, $D\big(\mathcal{U}^*, \mathcal{U}^{(t)}\big)$ is upper bounded by $2C\xi_1$, which gives the optimal statistical rate. Solving $t$ in this inequality, we obtain that $t = \widetilde{T} \geq \widetilde{T}_{\min}$, which is defined in (17).

**Analysis of the Initialization Stage:** Let $\overline{\boldsymbol{\Pi}}^{(t)} = 1/t \cdot \sum_{i=1}^{t} \boldsymbol{\Pi}^{(i)}$ where $\boldsymbol{\Pi}^{(i)}$ is defined in Algorithm 3. Let $\mathcal{U}^{(t)}$ be the $k$-dimensional subspace spanned by the top $k$ leading eigenvectors of $\overline{\boldsymbol{\Pi}}^{(t)}$.

**Theorem 3.** Under the same condition as in Theorem 1, the iterative sequence of $k$-dimensional subspaces $\mathcal{U}^{(0)}, \mathcal{U}^{(1)}, \ldots, \mathcal{U}^{(t)}, \ldots$ satisfies

$$D\big(\mathcal{U}^*, \mathcal{U}^{(t)}\big) \leq \underbrace{\zeta_1}_{\text{Statistical Error}} + \underbrace{\zeta_2 \cdot 1/\sqrt{t}}_{\text{Optimization Error}} \qquad (19)$$

with high probability. Here $\zeta_1$ and $\zeta_2$ are defined in (15).

In Theorem 3 the optimization error term decays to zero at the rate of $1/\sqrt{t}$. Note that $\zeta_2$ increases with $d$ at the rate of $\sqrt{d} \cdot (\log d)^{1/4}$. That is to say, computationally convex relaxation is less efficient than sparse orthogonal iteration pursuit, which justifies the early stopping of ADMM. To ensure $\mathcal{U}^{(T)}$ enters the basin of attraction, we need $\zeta_1 + \zeta_2/\sqrt{T} \leq R$. Solving $T$ gives $T \geq T_{\min}$ where $T_{\min}$ is defined in (17). The proof of Theorem 3 is a nontrivial combination of optimization and statistical analysis under the variational inequality framework, which is provided in the extended version [29] of this paper with detail.

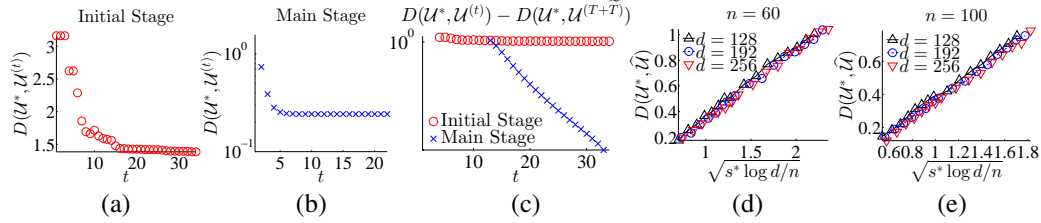

Figure 2: An Illustration of main results. See §5 for detailed experiment settings and the interpretation.

Table 1: A comparison of subspace estimation error with existing sparse PCA procedures. The error is measured by $D(\mathcal{U}^*, \widehat{\mathcal{U}})$ defined in (6). Standard deviations are provided in the parentheses.

| Procedure | $D(\mathcal{U}^*, \widehat{\mathcal{U}})$ for Setting (i) | $D(\mathcal{U}^*, \widehat{\mathcal{U}})$ for Setting (ii) |
|---|---|---|
| **Our Procedure** | **0.32** (0.0067) | **0.064** (0.00016) |
| Convex Relaxation [13] | 1.62 (0.0398) | 0.57 (0.021) |
| TPower [11] + Deflation Method [24] | 1.15 (0.1336) | 0.01 (0.00042) |
| GPower [10] + Deflation Method [24] | 1.84 (0.0226) | 1.75 (0.029) |
| PathSPCA [8] + Deflation Method [24] | 2.12 (0.0226) | 2.10 (0.018) |

(i): $d = 200$, $s = 10$, $k = 5$, $n = 50$, $\mathbf{\Sigma}$'s eigenvalues are $\{100, 100, 100, 100, 4, 1, \ldots, 1\}$;
(ii): The same as (i) except $n = 100$, $\mathbf{\Sigma}$'s eigenvalues are $\{300, 240, 180, 120, 60, 1, \ldots, 1\}$.

## 5 Numerical Results

Figure 2 illustrates the main theoretical results. For (a)-(c), we set $d=200$, $s^*=10$, $k=5$, $n=100$, and $\mathbf{\Sigma}$'s eigenvalues are $\{100, 100, 100, 100, 10, 1, \ldots, 1\}$. In detail, (a) illustrates the $1/\sqrt{t}$ decay of optimization error at the initialization stage; (b) illustrates the decay of the total estimation error (in log-scale) at the main stage; (c) illustrates the basin of attraction phenomenon, as well as the geometric decay of optimization error (in log-scale) of sparse orthogonal iteration pursuit as characterized in §4. For (d) and (e), the eigenstructure is the same, while $d$, $n$ and $s^*$ take multiple values. They show that the theoretical $\sqrt{s^* \log d/n}$ statistical rate of our estimator is tight in practice.

In Table 1, we compare the subspace error of our procedure with existing methods, where all except our procedure and convex relaxation [13] leverage the deflation method [24] for subspace estimation with $k > 1$. We consider two settings: Setting (i) is more challenging than setting (ii), since the top $k$ eigenvalues of $\mathbf{\Sigma}$ are not distinct, the eigengap is small and the sample size is smaller. Our procedure significantly outperforms other existing methods on subspace recovery in both settings.

**Acknowledgement:** This research is partially supported by the grants NSF IIS1408910, NSF IIS1332109, NIH R01MH102339, NIH R01GM083084, and NIH R01HG06841.

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
