[Reviews · NeurIPS 2014]

Submitted by Assigned_Reviewer_15

In the paper, a sparse PCA algorithm was studied. At the beginning, the authors introduced existing methods and their
drawbacks in terms of both statistical and computational properties. Then, a two stage algorithm was proposed to
overcome the drawbacks. In the first stage of the algorithm, a computationally efficient but statistically inefficient
method based on the ADMM is used to obtain a preliminary solution. In the second step, the algorithm called sparse
orthogonal iteration pursuit is used to improve the solution obtained in the first stage. Also, some theoretical
properties of the estimator were investigated. Numerical experiments showed the usefulness of the proposed methods
compared to some existing methods.

This paper is well written, though constants are very dense. Especially, theoretical studies include some remarkable
results such as the minimax-optimality of the algorithm, the blessing of large sample size, and analysis for non-Gauss
or dependent data.

Though I could not read the supplementary of the paper in such a short view period, I'll try to check it out in the
remaining review period, and give some comments.
Summary: This paper is well written, and includes some remarkable theoretical results.

Submitted by Assigned_Reviewer_21

In this paper, a two-stage sparse PCA procedure was proposed, which involves
a novel algorithm called orthogonal iteration pursuit. Under an integrated
analytic framework, the computational and statistical performance of this
two-stage procedure was characterized. Interestingly, the final principal
subspace estimator was shown to be able to achieve the minimax-optimal
convergence rate.
Summary: The paper seems very interesting and the results seems new. However,
I am not familiar with the literature of analysing sparse PCA
algorithms and not fully sure about the significance of the results.

Submitted by Assigned_Reviewer_30

The claim of this paper is impressive. They have a polynomial time algorithm for sparce PCA that is minimax-optimal. The result seems to be true and the authors offer a good background of previous work. The numerical results is not so developed but one can't have everything in 8 pages.

The paper is of good quality and the work is original.

The biggest problem I have with this paper is the authors did not clearly state when their method will not work. I understand that there has to be eigenvalue separation but is that all that is needed. Also the eigenvalue separation does enter the rate of the method, of course this is fine in the minimax analysis but it would be good to know how this scales. Also are there any complexity assumptions such as P \neq NP or properties of reductions for the computational claims or is it just a calculation of the number of operations. Two other points: 1) some intuition as to how the second algorithm is faster in falling into the right part of the basin of attraction that iterating the first part (I remember there were comments on this but they were diffuse). Any idea of how to select k ?
Summary: The claim of this paper is impressive. They have a polynomial time algorithm for sparce PCA that is minimax-optimal.
Author Feedback
Author rebuttal: We thank the reviewers for their very helpful comments. Here we address the raised concerns and will revise the final version accordingly.

Reviewer 1:

We thank the reviewer for the valuable suggestions. We agree that the constants are dense. In the final version, we will present these constants in a more compact way. Meanwhile, there are several crucial quantities e.g., \gamma, that cannot be merged into absolute constants, since these quantities might also scale with the sparsity level s^*, dimension d as well as sample size n, and hence affect the final statistical and computational rates of convergence. Thus it is necessary to explicitly involve these quantities in the final results. For these quantities, we will provide more intuition in the final version. Also, we would be very glad to hear more comments and feedbacks on the supplementary material from the reviewer.

Reviewer 2:

We thank the reviewer for the thoughtful comments. In short, our major contribution is a novel two-stage computational framework that efficiently attains the minimax-optimal principal subspace estimator, without relying on the spiked covariance assumption or the Gaussian assumption. Meanwhile, we develop new analytic techniques for theoretical justification. As suggested by the reviewer, in the final version we will present the related work in a more detailed manner to illustrate the significance of our main results.

Reviewer 3:

We thank the reviewer for the insightful comments. Regarding the settings where our procedure doesn't work, we have the following comments:

1. When the sample size is not sufficiently large, or the eigengap is very close to zero, e.g., in (15) the eigengap decays to zero at the rate of (\lambda_1 s^* \sqrt{log d/n}), the statistical error of the convex relaxation stage is large, such that \zeta_1 is greater than R in (17). Consequently, even if the initialization stage takes an infinite number of iterations, the iterative sequence of the convex relaxation stage is not guaranteed to fall into the basin of attraction for the main stage. In this case, our procedure will possibly fail (although not necessarily), which will be illustrated with an extra numerical result in the final version.

2. Suppose the initial estimator obtained from the initialization stage turns out to be good. For example, even with random initialization, we still have the chance to obtain a desired initial estimator for the main stage, although with a probability close to zero. In this situation, if the sample size n is not large enough, or the eigenvalue decreases too quickly to zero, as shown in the definition of the final statistical error \xi_1 in (16), then the final estimator \hat{\mathcal{U}} is not consistent. Moreover, in this case, the minimax lower bound suggests that no procedure can produce consistent estimator in this setting.

3. The above two settings only characterize the situation where our procedure fails to provably obtain statistically consistent estimator. From a computational point of view, it is interesting to further investigate when it is impossible to attain the optimal estimator in polynomial time. In parallel to our positive result, the negative result in Berthet and Rigollet 2013 [20] shows that, under the assumption that a certain type of planted clique problems are intractable to compute, it is impossible to compute the optimal estimator when the eigengap decreases to zero at the rate of (\sqrt{s^* \log d/n}). Meanwhile, as discussed in Section 4, our positive result doesn't allow the eigengap to decrease to zero at such a fast rate, and thus is not subject to the limit of their negative result. Also, since ours is a positive result, we doesn't rely on the P \neq NP assumption or the planted clique assumption. We will provide more discussion on this relationship between positive and negative results of sparse PCA in the final version. It is also interesting to investigate the sharp phase transition between these two regimes.

Regarding the two other points raised by the reviewer, we have the following explanations:

1. The second algorithm is faster in two aspects. From the aspect of iteration complexity, the first algorithm converges at the rate of 1/\sqrt{t} in terms of optimization error, while the second algorithm converges at a geometric rate. From the aspect of computational cost within each iteration, a single iteration of the first algorithm is more expensive than that of the second algorithm (roughly by an order of dimension d, which is assumed to be very large). Hence, the second algorithm is faster. Thus, when the iterative sequence enters the basin of attraction, we should switch to the second algorithm to reduce the computational cost.

2. Currently we assume that the dimension of subspace k is known in advance, e.g., in the application of population genetics, the number of population groups is prefixed, e.g., we have asian, african, european, ... . If k is unknown, the choice of k might differ case by case. For example, if we assume the population covariance matrix is only rank k, i.e., the rest of its eigenvalues are zero. Then one possible approach for determining k is to construct some test on whether a certain eigenvalue of the population covariance matrix is zero. See ''Some limit theorems for the eigenvalues of a sample covariance matrix'' by Dag Jonsson for an example. It is interesting to investigate the behavior of this test in high dimensional settings. Meanwhile, in other settings, one might choose k by examining the variance explained by principal components, or simply investigate the regularization path for k =1, 2, 3, ..., as in the Lasso setting, in which we are interested in the path of the regularization parameter \lambda.